# Applying Biofloc Technology in the Culture of *Mugil cephalus* in Subtropical Conditions: Effects on Water Quality and Growth Parameters



Sara Garcés and Gabriele Lara *

Escuela de Ciencias del Mar, Pontificia Universidad Católica de Valparaíso, Avenida Universidad 330, Valparaíso 2373223, Chile; sara.chaverra@gmail.com
* Correspondence: gabriele.rodrigues@pucv.cl

**Abstract:** *Mugil cephalus* is an estuarine species that has been shown to acclimate to a diversity of environmental conditions being a good candidate to diversify aquaculture. The objective of this study was to evaluate the growth and water quality of *M. cephalus* juveniles in a biofloc technology system (BFT). Over a period of 87 days, mullet juveniles (117.36 ± 6.48 g) were reared in two experimental conditions: water exchange (control group) (50% three times a week) and BFT (C:N ratio of 15:1). *M. cephalus* juveniles were stocked at a density of 7.2 kg·m$^{-3}$. Water quality parameters were monitored periodically. Fish were fed with a balanced artificial feed for marine fish (2% of the total biomass). The water quality parameters were similar between the treatments, with the exception of pH, alkalinity, TSS, and N-NO$_3$ ($p < 0.05$). Zootechnical parameters WG, DWG, BG, SGR, condition factor, and survival did not show significant differences ($p > 0.05$). The biomass gain was higher in BFT ($p < 0.05$). Water quality variables did not appear to influence mullet survival under culture conditions. Mullet is a species that can be reared in BFT without compromising productive performance and having a significant saving of water.

**Keywords:** BFT system; Mugilidae; water quality; fish culture; low temperatures

**Key Contribution:** Biofloc technology can constitute a sustainable strategy for aquaculture with new species in subtropical zones.





## 1. Introduction

The flathead mullet, *Mugil cephalus*, is a migratory and cosmopolitan species, occurring in marine, freshwater, and estuarine environments [1]. *M. cephalus* is the most widespread species among the family Mugilidae. Along with other Mugilidae species, *M. cephalus* has become a popular hydrobiological resource for fisheries and aquaculture in many parts of the world [2–4]. Its spawning occurs in the sea, and then fishes migrate to estuarine and brackish environments to carry out breeding [5,6]. It can be described as an omnivorous and zooplanktophagous species [7]. According to Whitfield et al. [5] in the first stages of development, flathead mullet is planktophage, whereas, in the adult stage, it is considered detritivore, foraging at the base of the food web in all life stages. On the other hand, the stomach content of fishes in natural environment shows that this species has a preference for diatoms, dinoflagellates, silicoflagellates, tintinnids, and copepods [8]. Jamabo and Maduako [7] observed that, in addition to plant remains, other food items consumed are organic matter, mud, annelids, crustaceans, fish remains, and insect remains.

At present, flathead mullet is reared in some countries such as Egypt, Senegal, Italy, and other Mediterranean countries [9–11]. Fry used in commercial aquaculture are collected from the natural environment and subsequently stocked into lagoons, lakes, or ponds in fresh or brackish water, mostly grown in semi-intensive polyculture systems [5,11].

In Chile, the landing of mullet has been increasing gradually since the last 10 years, mostly in small-scale fisheries [12]. In many countries, it is a species with a growing interest for fishing because its gonads have a high commercial value. Consequently, both overfishing and habitat alteration are resulting in major declines in *M. cephalus* stocks around the world [5]. To avoid reducing the wild stock, it is necessary to generate strategies that promote the management of its culture using sustainable technologies [13].

Various studies have been carried out focused on the trophic ecology, migratory movements, taxonomy, genetics, and biology of the species [1,3,6,8–10,13–15]. Under culture conditions, however, the studies present results for a wide variety of rearing systems in different conditions, structures, and intensities (extensive, semi-intensive, closed and open systems, monoculture, polyculture, using ponds or tanks) [16–25].

In recent decades, with the growth of the aquaculture industry worldwide, the need to migrate toward environmentally friendly technologies has been highlighted, using resources more rationally and minimizing potential negative effects on the environment [26]. The biofloc technology system (BFT) is an emerging technology that meets the requirements for the industry to continue growing, with a view to a more sustainable aquaculture [27–29]. Through increasing the C:N ratio (10–20:1) in the water [30–33], a variety of microorganisms (bacteria, microalgae, protozoa, rotifers, and nematodes, among others) develop, forming bioflocs [28,34]. These aggregates contribute to the improvement of the system's water quality, as well as serving supplementary feed to reared organisms [27,35,36]. In this way, the need for water exchange is reduced and artificial feed can be spared, improving feed efficiency. Due to its characteristics, the BFT system is also considered more biosecure, increasing the immunity of the organisms and not using antibiotics during the cycle. Currently, shrimp and tilapia are the main species commercially farmed in this system [28,36–39]; however, with the aim of diversifying aquaculture, new studies have been carried out to find new species that may be suitable to be reared in BFT, taking advantage of their productive benefits.

Due to its feeding habits, the flathead mullet can be considered a promising species to be reared in BFT since it can take advantage of the microbial aggregates that are found in suspension in the water. Furthermore, being an estuarine and cosmopolitan species, it is considered a species that has a wide tolerance to water quality parameters, as well as being able to live in environments with high turbidity [5,18,21,40–43], an intrinsic characteristic of the BFT system. In other Mugilidae species, it has been shown that fishes can efficiently remove the excess of suspended solids in BFT and transform them into biomass [36,39,44,45]. In this way, the microbial flocs responsible for improving water quality can be used as a source of food supplement and promote the growth of reared organisms [27,28,35,37,46].

Knowing the need to increase the number of species with the potential to be cultivated in sustainable culture systems, the present study aims to evaluate the growth performance of *M. cephalus* juveniles and water quality in a BFT culture system.

## 2. Materials and Methods

### 2.1. Ethics

The experiment conducted in the present study was approved by the Bioethics and Biosafety Committee of the Pontificia Universidad Católica de Valparaíso (Code: BIOPUCV-BA 465-2021). Russell and Burch's 3Rs criteria were applied. Constant supervision of the behavior of the animals, monitoring, and permanent control of water quality were carried out in order to maintain the animal welfare. Maintenance of water quality was carried out according to the environmental requirements of estuarine species. All procedures involving manipulation (biometrics) were performed using anesthesia to minimize stress and suffering of fishes.

### 2.2. Area of Study

Juvenile flathead mullets were collected from the "Río Maipo" estuary with the support of artisanal fishermen from the province of San Antonio, Chile (33°37′10.4″ S

71°37′22.4″ W). The fish were transferred to a tank with oxygen supplement and water from the same estuary and transported to the Center of Investigation on Sustainable Aquaculture (Centro de Investigación en Acuicultura Sustentable—CIAS) of the School of Marine Sciences at the Pontificia Universidad Católica de Valparaíso. During transport, temperature, dissolved oxygen, and pH were constantly monitored, and the transportation to the facilities lasted approximately 90 min.

### 2.3. Culture Conditions

Once at the research center facilities, the fish were acclimatized to laboratory conditions. During this stage, 30–50% water exchanges were performed three times a week, in order to maintain the water quality parameters in the recommended levels for estuarine fishes. Gradually the salinity was adjusted to 30 from 15 ppt, and all fish were acclimated for 60 days after arrival in the laboratory before the experiment started to compensate for the effects of capture, handling, feeding, transport, and water quality conditions.

After the acclimation period, fish were randomly distributed to each experimental unit at a density of 7.2 kg·m$^{-3}$. Circular tanks with a volume of 250 L with permanent aeration supplied by a blower (air compressor) (1HP) and diffuser stones were used to carry out the study.

### 2.4. Experimental Design

A total of 90 *M. cephalus* juveniles with an average weight of 117.36 ± 6.48 g were randomly distributed in tanks (15 fish per tank), where two treatments were analyzed: water exchange (Control) and BFT. The control group consisted of the culture of juveniles with water exchange at a rate of 50% of the total volume of tanks three times a week and no organic carbon addition. In control, the C:N ratio used was an hypothetical value, considering the composition of artificial feed (C: 240 g of feed × 0.9 (90% dry matter) × 0.7 (30% of fish assimilation or 70% of waste that remains in water) × 0.5 (carbon content of the feed is ~50% based on dry matter) = 75.6 g of C; N: 240 g of feed × 0.9 (90% dry matter) × 0.7 (30% of fish assimilation or 70% of waste that remains in water) × 0.475 (47.5% crude protein content of feed)/6.25 (constant) = 11.49 g of N). The calculations resulted in a C:N ratio of approximately 6.58:1; this ratio was not measured during the study. In BFT, the culture was evaluated in a biofloc system with zero water exchange and addition of "chancaca" (see composition specification in Section 2.5) as a carbon source to maintain the C:N ratio in 15:1 according to [47,48]. Each treatment had three replicates and a total of six experimental units. The study lasted 87 days.

### 2.5. Biofloc Maturation

Prior to the start of the study, biofloc maturation was promoted to enhance the growth of microorganisms of the aggregates. In a 180 L useful volume plastic tank with permanent aeration, 150 L of water from the mullet culture and uneaten feed were added. According to the ammonium concentration measured every day, commercial sugar cane extract in bar (solid) "chancaca" grated (each 100 g—energy: 377 kcal; protein 0.3 g; carbohydrates 93 g; total sugar 93 g; Aconcagua Foods, San Bernardo, Chile) with 36% carbon according to [49] was added as a carbon source. The maturation continued until a decrease in the TAN ($NH_3$ + $NH_4^+$) and nitrite values and an increase in the nitrate concentration were observed. Subsequently, 15 L of the mature biofloc (an inoculum of 6% of total volume of the tanks) was added to the experimental units of BFT treatment.

### 2.6. Water Quality Parameters

Temperature, DO (Multiparameter probe HACH HQ40; Iowa City, IA, USA) and salinity were measured daily (a.m. and p.m.), and pH (pH meter Ohaus STARTER 3100M; Parsippany, NJ, USA) was measured every two days (a.m. and p.m.). pH correction was made using calcium hydroxide (Ca(OH$_2$)) according to [50]. The nitrogen compounds and alkalinity were analyzed in spectrophotometer (HACH DR3900; Iowa City, IA, USA)

using specific kits from the same manufacturer as follows: total ammonia nitrogen (TAN; $NH_3 + NH_4^+$) (Nessler Kit, $NH_3$-N HACH), nitrite ($NO_2$-N) (kit HACH NitriVer® 3 Nitrite Reagent Powder Pillows, 10 mL mg/L $NO_2$-N. 2107169), nitrate ($NO_3$-N) (kit HACH NitraVer® 5 Nitrate Reagent Powder Pillows 10 mL mg/L $NO_3$-N. 2106169), alkalinity (Kit HACH Alkalinity (Total) TNTplus Vial Test (25–400 mg/L $CaCO_3$), and TSS (APHA, 2005) once a week.

### 2.7. Feed

The fish were fed with a balanced artificial feed for marine fish (47.5% CP, 19% lipids, 19.5 MJ·kg$^{-1}$ energy) twice a day (a.m. and p.m.) at a rate of 2% of the total biomass (kg), according to the feeding protocol used by [16]. The feed was adjusted each month according to the results of the biometrics.

### 2.8. Growth Performance

Every 30 days, the total number of fish in each tank was sampled to monitor body weight (g) and standard length SL (cm). The individuals were weighed using an electronic scale (JADEVER-JWE-6K; Fuzhou, China) and measured with a graduated ruler. Weight gain (g) was calculated as WG = Wf − Wi, and daily weight gain (g) was calculated as DWG = WG/time, while the specific growth rate (SGR %/day) was calculated using the equation SGR (%/day) = [(lnPf − lnPi) × 100]/time (days), where Wf is the final weight and Wi is the initial weight (g). The condition factor (K) was calculated as K = (100 × Wf)/SL3), where Wf is the final weight (g), and SL is the standard length (cm). Survival was calculated as survival (%) = (final N° of fish harvested/initial N° of fish stocked) × 100.

Before each biometry, the fish were subjected to a previous fasting. For the weighing and measurement of each tank, fish were extracted one by one with a fishing mesh and submerged in a bucket with anesthesia (0.24 g/L Tricaine methanesulfonate 80%, Centrovet Ltd., Santiago, Chile). At the end of the measurement, fish were placed in a tank with freshwater and aeration for their subsequent recovery.

Biometrics were performed every 30 days in order of evaluating weight gain, daily weight gain, specific growth rate, and condition factor. At the end of the study, fish were individually counted to evaluate the survival (%).

### 2.9. Statistical Analysis

Data were expressed as the mean ± standard deviation. For all the data of interest, the assumptions of normality (Shapiro–Wilk test) and homogeneity of variance (Levene's test) were verified. To identify the differences in weight gain, daily weight gain, biomass gain, SGR, and condition factor (k), as well as the differences in water quality variables between water exchange and BFT, a t-test was performed with a significance level of 0.05. The data that did not meet the assumptions were analyzed by means of a non-parametric Mann–Whitney U test. All the data were analyzed using Rstudio.

## 3. Results and Discussion

### 3.1. Water Quality Parameters

The water quality parameters (Table 1) did not present statistically significant differences ($p > 0.05$) between the treatments, with the exception of pH, alkalinity, TSS, and N-$NO_3$ ($p < 0.05$). The three parameters mentioned are directly related to the conditions of the BFT system, in which the formation, aggregation, and metabolism of microbial communities, especially nitrifying autotrophic and heterotrophic bacteria, consume alkalinity, reducing pH, increasing TSS, and transforming ammonium into nitrate, due to the nitrification process [47,51–54].

**Table 1.** Mean water quality parameters in the culture of *M. cephalus* in treatments with water exchange and biofloc for 87 days.

| Treatment/Parameter | Water Exchange | BFT |
|---|---|---|
| T (°C) | 15.91 ± 0.29 | 15.48 ± 0.10 |
| D.O. (mg·L$^{-1}$) | 8.37 ± 0.10 | 8.41 ± 0.04 |
| pH | 7.86 ± 0.06 [a] | 7.23 ± 0.18 [b] |
| Alkalinity (mg CaCO$_3$·L$^{-1}$) | 434.25 ± 15.31 [a] | 369.61 ± 94.15 [b] |
| Sal (ppt) | 13.88 ± 0.17 | 14.50 ± 0.13 |
| TAN (mg·L$^{-1}$) | 2.13 ± 0.44 | 2.07 ± 0.33 |
| NO$_2$ (mg·L$^{-1}$) | 0.67 ± 0.27 | 0.91 ± 0.76 |
| NO$_3$ (mg·L$^{-1}$) | 5.64 ± 0.63 [a] | 17.34 ± 1.52 [b] |
| TSS (mg·L$^{-1}$) | 150 ± 23.6 [a] | 370 ± 129.4 [b] |

Data are expressed as the mean ± standard deviation. Values with different letters express significant statistical differences ($p < 0.05$).

Estuarine fish are generally characterized by having a wide tolerance to variable ranges of salinity, dissolved oxygen, pH, temperature, and nitrogenous compounds, due to the natural variations that occur in their original ecosystems [5]. In the present study, water quality variables did not appear to influence mullet survival under culture conditions. However, Kibenge [55] reports that the ideal temperature for the growth of the species is 20–26 °C, a higher temperature than recorded in the present study. According to Prakoso [56], culturing grey mullets in brackish water could be applied at 25 °C to optimize aquaculture of this species. The low temperature registered in the present study explains in part the low growth of *M. cephalus* during the experiment, despite the fact that studies with this species show very wide temperature ranges in its culture. The pH was stable during the development of the study and was lower in the treatment with biofloc ($p < 0.05$). Similar results were observed by Hoang et al. [21], Klas et al. [40], and Nguyen et al. [25]. Concomitantly, alkalinity (434–369 mgCaCO$_3$·L$^{-1}$) was adequate to maintain the buffering capacity of the culture water in both systems; however, it was lower ($p < 0.05$) in the treatment with biofloc. Ebeling et al. [47] reported that a significant amount of alkalinity is consumed by autotrophic bacteria in closed systems, which explains the lower concentrations of alkalinity in the present study. Da Rocha et al. [44] found alkalinity values above 200 mg·L$^{-1}$ for *M. hospes* cultured in a BFT system. Similar values were observed in this study. In cultures of white shrimp (*Litopenaeus vannamei*) and Nile tilapia (*Oreochromis niloticus*) in biofloc technology, lower alkalinity values have been reported [25,31,36,38,45]. According to Ebeling et al. [47], alkalinity (mg CaCO$_3$·L$^{-1}$) in this study can be considered adequate for a biofloc system.

With respect to TAN, concentrations did not appear to negatively affect the survival of mullet. Similar values (2.17 mg·L$^{-1}$) were observed by Rocha et al. [44] for *Mugil hospes* in biofloc culture and are considered safe according to values reported by Sampaio et al. [57], who observed that a concentration of 2.07 mg·L$^{-1}$ at a salinity of 15 ppt is safe for juvenile *Mugil platanus*. Moreover, TAN concentrations did not show statistically significant differences between treatments ($p > 0.05$), indicating that TAN concentrations were maintained in adequate levels either by water exchange in control or by the activity of bacteria in the BFT treatment. In contrast, Vinatea et al. [58] observed higher levels of TAN in the culture of mullet (*M. cephalus*) fingerlings in BFT, which the authors attributed to the high level of protein in the diet (54% CP). The same authors indicated that the survival observed (85%) was negatively influenced by TAN when using biofloc.

Nitrite concentrations did not present significant differences between treatments ($p > 0.05$), indicating that microbial communities of biofloc were able to convert TAN into nitrite. In the same way, nitrite concentration was maintained at the safe level recommended by Sampaio et al. [57], with a salinity of 15 ppt. for other Mugilidae species. These authors suggested that special care should be taken with the acclimatization of wild estuarine species to different culture conditions but related that mugil species are highly tolerant

to nitrogen compound concentrations. In the case of *M. platanus* juveniles, non-ionized ammonium and nitrite toxicity is higher in freshwater compared to seawater.

Nitrate is a nitrogen compound that is less toxic for fish than ammonia or nitrite [59]. However, in closed aquaculture systems, such as the BFT system, it may accumulate in the water due to limited water exchange or high stocking density and cause negative effects on fish growth and survival. According to Presa et al. [60], who studied the acute toxicity of nitrate on *Mugil liza* in freshwater, it is recommended to maintain the concentrations of this compound under 52.66 mg·L$^{-1}$ in order to avoid negative effects on the physiological homeostasis of fishes. In the same way, Poersch et al. [61] found that a concentration of 152.2 mg·L$^{-1}$ does not compromise the performance of *M. platanus* fingerlings and can be considered as safe. In the present study, nitrate concentrations were maintained below these levels in brackish water, which is also negatively related to nitrate toxicity. In summary, the use of a mature biofloc (with low concentrations of TAN and NO$_2$ and high concentrations of NO$_3$) at the starting point of the experiment helped in maintaining stable concentrations of these compounds throughout the experiment. These concentrations did not fluctuate expressively during the study and did not show significant differences over time.

As a product of the formation of microbial aggregates, TSS was significantly higher in BFT ($p < 0.05$). In this treatment, mean TSS concentration reached $370 \pm 129$ mg·L$^{-1}$, which is according to the recommended levels for biofloc culture. Rocha et al. [44] reported TSS values ranging from 310.67 to 785.33 mg·L$^{-1}$ for the culture of *M. hospes* in biofloc, much higher than those obtained in this study. Since the TSS concentrations did not reach high values, it was not necessary to control TSS with sedimentation/clarification processes. It is known that mullet species can control the TSS concentrations in BFT systems, as studies of Legarda et al. [45] and Holanda et al. [36] observed a significant reduction in TSS in the integrated culture of *M. liza* and white shrimp in biofloc technology compared to shrimp monoculture. These authors related the reduction in TSS to the consumption of biofloc by mullet species. In the same way, Madrid et al. [24] pointed out that mullet juveniles have the ability to use nutrient residues, and that biofloc can be considered a good food alternative. Similar results were found by Katz et al. [2], who suggested that mullets can be an excellent bioremediator since they are capable of reducing the sediment from the culture of fish in cages in short periods of time. Contrasting results were observed by Hoang et al. [62] in the polyculture of mullet and white shrimp in biofloc. These authors suggested that mullet is not very efficient at consuming the flocs; however, in this study, a low concentration of flocs was observed (accompanied by an increase in the concentration of ammonium and nitrite at the end of the experiment), which may indicate that the mullet juveniles may have consumed them (affecting the removal of nitrogenous compounds). In the present study, due to the fact that the TSS concentrations did not reach high values, it can be indicated that the fish exerted predation on the biofloc, contributing to the control of this parameter.

In conventional aquaculture systems, one of the commonly used methods to improve water quality is water exchange. In this study, the control treatment received a replacement of 1.125 L·week$^{-1}$, equivalent to 13.845 L during the experimental period. In treatment using BFT, 45 L·week$^{-1}$ was used, with a total of 552.6 L corresponding to the replacements made by evaporation and removal of waste from the drain. Hence, the culture of mullet in BFT presented a significant saving (92.3%) of the total water resource. These results show that BFT is a sustainable alternative to flowthrough aquaculture systems, saving water and maintaining water quality at suitable levels to fishes.

### 3.2. Growth Performance

The data of the productive performance parameters: WG, DWG, BG, SGR, condition factor K, and survival are presented in Table 2.

**Table 2.** Zootechnical performance parameters of *M. cephalus* in a system with water exchange and BFT during 87 days of the study.

| Parameter | Water Exchange | Biofloc |
| --- | --- | --- |
| Initial weight (g) | 117.11 ± 8.7 | 117.62 ± 7.1 |
| Final weight (g) | 124.82 ± 7.8 | 129.70 ± 7.2 |
| Weight gain (g) | 7.72 ± 1.5 | 11.48 ± 2.8 |
| Daily growth rate (g) | 0.09 ± 0.02 | 0.13 ± 0.03 |
| Initial biomass (Kg) | 1.83 ± 0.1 | 1.77 ± 0.1 |
| Final biomass (Kg) | 1.95 ± 0.1 | 1.95 ± 0.1 |
| Biomass gain (g) | 122.40 ± 34.7 [a] | 171.65 ± 43.8 [b] |
| Specific growth rate (%/day) | 0.07 ± 0.02 | 0.11 ± 0.03 |
| Initial condition factor (Ki) | 1.42 ± 0.06 | 1.49 ± 0.01 |
| Final condition factor (Kf) | 1.51 ± 0.04 | 1.54 ± 0.0001 |
| Survival (%) | 100 | 100 |

Data are expressed as the mean ± standard deviation. Values with different letters express significant statistical differences ($p < 0.05$).

The main growth parameters did not show significant differences between the treatments, which means that the BFT system did not affect the productive performance of the species. The only zootechnical parameter which presented a significant difference between treatments was the biomass gain, being higher in the BFT system ($p < 0.05$). Studies with *M. cephalus* and other mugilid species are variable according to the growth performance of mullet in different culture systems and applying different management strategies. Hoang et al. [62] obtained higher total production per tank for mullet fingerlings using BFT in co-culture with shrimp. These authors reported a higher DWG (1.31 ± 0.02 g) than observed in the present study. Vinatea et al. [58] observed a lower weight gain (1.66 g) and a higher SGR (1.87%/day) for mullet fingerlings in BFT. Biswas et al. [18] reported daily weight gains of 0.23 to 0.63 g for mullet fingerlings grown in soil ponds with and without fertilization. However, the same study presented much lower survival rates, ranging from 50% to 83.2%, while, in this study, the survival in both treatments was 100%. Hoang et al. [62] also reported a lower survival rate for *M. cephalus* (74–82%) in polyculture with white shrimp.

Similarly, Mehrim et al. [20] evaluated the performance of mullet (66 ± 1.2 g) in polyculture with tilapia, as well as each in monoculture. The highest values of GDP and SGR were observed when the mullet was cultured in polyculture with tilapia (2.25 ± 0.88 g and 141 ± 0.28%/day) and in polyculture with tilapia and *M. capito* (1.07 ± 0.10 g and 0.94 ± 0.08%/day), respectively. In contrast, the lowest values were obtained in monoculture (0.91 ± 0.13 g and 0.84 ± 0.08%/day). Likewise, Borges et al. [39] evaluated the performance of *M. liza* in co-culture with white shrimp and in monoculture using BFT, observing lower growth in monoculture. The authors suggested that the productive performance of Mugilidae can be considered low.

The artificial feed used in this study could have negatively influenced the performance of juveniles because the protein and lipid levels were above the recommended levels for Mugilidae at this stage of development [3]. These same authors point out that a crude protein level of 300–327 g·K$^{-1}$ is adequate for the culture of *M. cephalus* fingerlings and that higher dietary values do not allow obtaining greater growth or additional benefits. Additionally, during the development of this study it was observed that the feed offered was not always consumed in its entirety.

In the domestication of different animal species in captivity, it is expected that these organisms do not present a satisfactory growth in relation to others reproduced and adapted in the laboratory [63]. In the present study, low growth could have been related to some factors: low water temperatures, acclimatization of wild species to the artificial culture environment, use of an artificially formulated feed that possibly did not meet the nutritional requirements of the fish, and the greater size of the organisms when compared to other

published studies. These factors together may have compromised better growth of the fish in the trial.

## 4. Conclusions

According to the results obtained in this study, mullet is a species that can be reared in BFT systems without compromising productive performance and having a significant saving of water resources (92.3%). The culture of *M. cephalus* in BFT is a strategy that can contribute to the sustainability of aquaculture; however, it is necessary to continue carrying out studies that involve cultivation at different stages of development.

**Author Contributions:** Conceptualization, S.G. and G.L.; methodology, S.G. and G.L.; formal analysis, S.G. and G.L.; investigation, S.G. and G.L.; data curation, G.L.; writing—original draft preparation, S.G. and G.L.; supervision, S.G.; project administration, G.L.; funding acquisition, G.L. All authors have read and agreed to the published version of the manuscript.

**Funding:** This research was funded by the ANID + FONDEF Project ID21I10088 and PUCV Emerging Research DI project 2021 (039.338/2021).

**Institutional Review Board Statement:** The methods used for the development of this study were approved by the ethics and biosafety committee of the Pontificia Universidad Católica de Valparaíso. (CODE BIOPUCV-BA 465-2021).

**Data Availability Statement:** Data supporting the findings of this investigation will be made available upon request.

**Acknowledgments:** The development of this study was possible due to the collaboration of the School of Marine Sciences of the Pontificia Universidad Católica de Valparaíso and the CIAS Sustainable Aquaculture Research Center. The authors thank Juan Pablo Monsalve and Pablo Mejías for their technical assistance.

**Conflicts of Interest:** The authors declare no conflict of interest.

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
