# Peer review of "Applying Biofloc Technology in the Culture of Mugil cephalus in Subtropical Conditions: Effects on Water Quality and Growth Parameters"

_fishes, doi:10.3390/fishes8080420_

Round 1

Reviewer 1 Report

The research presents initial considerations for applying biofloc technology to farm Mugil cephalus, while also identifying the associated challenges and potentials. By exploring the application of this technology to cultivate Mugil, the study sheds light on its feasibility and provides a foundation for further investigation. The focus on the utilization of a low trophic species for aquaculture can help the industry to diversify its production and reduce the reliance on high trophic species. Despite that, a suboptimal growth performance of M. cephalus was observed, but the high survival rate suggests the potential utility of the biofloc system in cultivating mugil. Further research and optimization of the system parameters may lead to improved growth outcomes. The authors should consider a few suggestions to improve the quality of the study, as follows. Other small corrections/suggestions are presented in the file.

  1. Table 1 appears unnecessary since water quality measurement, including frequencies for certain parameters, is already detailed in section 2.7. To streamline the information, the author could simply incorporate equipment models and methods used (within brackets) directly into the text.

  1. Was TSS removal performed by the authors during the trial? If not, it is recommended to explicitly mention this in the text, to reinforce the suggestion of M. cephalus's ability to control TSS in the system.

  1. Did the nitrogenous compound exhibit any fluctuations during the cultivation period? Additionally, was the strategy of maturing the bioflocs and utilizing them as an inoculum efficient? It would be interesting for the authors to provide further elaboration on these topics.

  1. Authors should consider changing the term "ration" to "feed" in the text, which is more commonly used in the literature.

  1. On table 3, "Growth rate" should be replaced with "Weight gain”, since the data reported is given in grams, not a rate (g/variable).

Just minor corrections, highlighted on the file.

Author Response

Reviewer 1 

The research presents initial considerations for applying biofloc technology to farm Mugil cephalus, while also identifying the associated challenges and potentials. By exploring the application of this technology to cultivate Mugil, the study sheds light on its feasibility and provides a foundation for further investigation. The focus on the utilization of a low trophic species for aquaculture can help the industry to diversify its production and reduce the reliance on high trophic species. Despite that, a suboptimal growth performance of M. cephalus was observed, but the high survival rate suggests the potential utility of the biofloc system in cultivating mugil. Further research and optimization of the system parameters may lead to improved growth outcomes. The authors should consider a few suggestions to improve the quality of the study, as follows. Other small corrections/suggestions are presented in the file.

Response: Thank you for your revisions and suggestions. The corrections were made directly in the main text and the responses to the considerations are contested above and inserted/removed into the file.. 

Table 1 appears unnecessary since water quality measurement, including frequencies for certain parameters, is already detailed in section 2.7. To streamline the information, the author could simply incorporate equipment models and methods used (within brackets) directly into the text. 

Response: Modifications were made into text, Table 1 was removed.

 Was TSS removal performed by the authors during the trial? If not, it is recommended to explicitly mention this in the text, to reinforce the suggestion of M. cephalus's ability to control TSS in the system. 

Response:  It was not necessary to remove TSS in BFT because the concentration did not exceed 500 mg.L-1, the recommended value for cultures in BFT. In the water exchange treatment, periodic water exchanges prevented the accumulation of TSS. A sentence supporting this information was added to the text. 

Did the nitrogenous compound exhibit any fluctuations during the cultivation period? Additionally, was the strategy of maturing the bioflocs and utilizing them as an inoculum efficient? It would be interesting for the authors to provide further elaboration on these topics.

Response: The use of a mature biofloc (with low concentrations of TAN and NO2 and high concentrations of NO3) helped in maintaining stable concentrations of these compounds during all experimental time. These concentrations did not fluctuate expressively during the study, and did not show significant differences along time. 

Authors should consider changing the term "ration" to "feed" in the text, which is more commonly used in the literature. 

Response: The term is modified in the text.

On table 3, "Growth rate" should be replaced with "Weight gain”, since the data reported is given in grams, not a rate (g/variable). 

Response: The term is modified in the text.

Reviewer 2 Report

The paper “Applying biofloc technology in the culture of Mugil cephalus in subtropical conditions: effects on water quality and growth parameters” is presented clearly and orderly. The experiments were well conducted, the results well presented, and the conclusions were supported and integrated with other experiences. Although there are many reports about the benefits of biofloc technology, Mugil cephalus is a species that has been incorporated into cultivable species in recent years, mainly to protect natural populations. This adds interest to the research they present.

Author Response

Reviewer 2

The paper “Applying biofloc technology in the culture of Mugil cephalus in subtropical conditions: effects on water quality and growth parameters” is presented clearly and orderly. The experiments were well conducted, the results well presented, and the conclusions were supported and integrated with other experiences. Although there are many reports about the benefits of biofloc technology, Mugil cephalus is a species that has been incorporated into cultivable species in recent years, mainly to protect natural populations. This adds interest to the research they present.

Response: We appreciate your consideration. M. cephalus is a species with high potential for aquaculture diversification and we are very excited to present more information about BFT culture with new species. 

Reviewer 3 Report

The manuscript presents considerable interest and results are promising. However, some essential details are missing in the manuscript, especially in the materials and methods section. 

Introduction and discussion sections need to be explored and revised. My suggestion is to accept this manuscript with major revision.

Major comment:

1. Please add commercial and technical details of all used products/equipment, in order to allow reproducibility of the study.

2. What about the C:N ratio of T1 group? This information should be added and the concept better explained to the reader.

3. In many parts of the manuscript several Latin fish names are not written in italics, some grammatical errors must be revised and some acronyms/abbreviations must be defined the first time they appear in each of three sections: the abstract; the main text; the first figure or table (see journal instructions for authors). 

Also detailed information on the composition of the "chancaca" must be added to the manuscript.

4. Why don’t you name the two experimental groups control (only water exchange) and treated (BFT)? 

5. Some parts of the introduction should be moved in order to make the reading easier. e.g. sentence in lines 40-41 should be moved to line 30, before the spawning statement; or before describing the situation in Chile (L42), insert the statement made on lines 48-51.

6. Please specify the adopted measures to minimize animal suffering in paragraph 2.1.

7. Experimental design: please specify with more details how each unit was designed. You write that 92 fish in total were used, divided into 6 experimental units. how many fish per experimental unit? It’s not clear, please write it more clearly.

8. Line 118: C/N ratio 15:1 according to ref. 47 and 48. Line 126: C/N ratio 15:1 according to ref. 49. Please check.

9. Fish feed: please add the energy content of the feed and some reference on the feeding practice used.

10. Line 151: Tricaine 80%? please add the exact concentration used for fish sedation.

After this round, minor comments will be evaluated.

 Minor editing of English language required

Author Response

Reviewer 3 

The manuscript presents considerable interest and results are promising. However, some essential details are missing in the manuscript, especially in the materials and methods section. Introduction and discussion sections need to be explored and revised. My suggestion is to accept this manuscript with major revision.

Response: thank you for the deep review of the manuscript. We hope that the modifications made in the text and the responses to the comments improved the quality of the article. 

Major comment:

  1. Please add commercial and technical details of all used products/equipment, in order to allow reproducibility of the study.

Response: The commercial and technical details of the products/equipment were provided in text. If it is necessary more information let us know. 

  1. What about the C:N ratio of the T1 group? This information should be added and the concept better explained to the reader. 

Response: Both carbon and nitrogen are elements present in food. In this sense, in the control treatment the C:N ratio of the system is given by the amount of carbon and nitrogen in the diet. In this treatment, there was no external addition of carbohydrates other than what enters the system through the artificial food. Therefore, in this study when using a food with a composition of 47.5% crude protein, a C/N ratio of 6.58:1 would be generated. However, these values are hypothetical, given that the C:N ratio was not measured in this study. We elucidated this point in the material and methods and provide the summary of calculations above (if is necessary to add the calculations to the main text let us know): 

Summary of calculations

C: 240g of feed × 0.9 (90% dry matter) × 0.7 (30% of fish assimilation or 70% of waste that remains in water)*0,5 (carbon content of the feed is ~50% based on dry matter) = 75,6 g of C 

N: 240g of feed × 0.9 (90% dry matter) × 0.7 (30% of fish assimilation or 70% of waste that remains in water) × 0.475 (47.5% crude protein content of feed)/6.25 (constant) = 11.49g of N. The results indicated a ~6.58:1 C:N ratio of feed 

  1. In many parts of the manuscript several Latin fish names are not written in italics, some grammatical errors must be revised and some acronyms/abbreviations must be defined the first time they appear in each of three sections: the abstract; the main text; the first figure or table (see journal instructions for authors). Also detailed information on the composition of the "chancaca" must be added to the manuscript. 

Response: modifications are made in the main text. We believe that some words that were in italics in our version were changed to normal lettering in the journal's document format. We have included all words in italics again.

  1. Why don’t you name the two experimental groups control (only water exchange) and treated (BFT)?

Response: We modified the treatments to Water Exchange and BFT.  

  1. Some parts of the introduction should be moved in order to make the reading easier. e.g. sentence in lines 40-41 should be moved to line 30, before the spawning statement; or before describing the situation in Chile (L42), insert the statement made on lines 48-51. 

Response: The suggested reorder of the text was made. 

  1. Please specify the adopted measures to minimize animal suffering in paragraph 2.1.

Russell and Burch's 3Rs criteria were applied. Constant supervision of the behavior of the animals, monitoring and permanent control of water quality were carried out. Maintenance of water quality was carried out according to the environmental requirements of estuarine species. All procedures involving manipulation (biometrics) were performed using anesthesia to minimize stress.

  1. Experimental design: please specify with more details how each unit was designed. You write that 92 fish in total were used, divided into 6 experimental units. How many fish per experimental unit? It’s not clear, please write it more clearly. 

Response: We made a mistake, the initial number of fish was 92, however two fish were removed for other analysis which is not presented in this article. A number of 90 fish were actually used (15 fish per tank) and  this data was corrected in the text to avoid misunderstandings. 

  1. Line 118: C/N ratio 15:1 according to ref. 47 and 48. Line 126: C/N ratio 15:1 according to ref. 49. Please check. This was a mistake citation, due to author 49 who described carbon % of chancaca. 

Response: The references were corrected in the text.

  1. Fish feed: please add the energy content of the feed and some reference on the feeding practice used.

Response: Reference of feeding practice was added in the methodology. The energy content of the feed was approximately 22 MJ.Kg-1. This data was provided by the feed manufacturer. 

  1. Line 151: Tricaine 80%? please add the exact concentration used for fish sedation. 

Response: A dose of 0.24 g/L  of tricaine was used for fish sedation. The dose used was added in the text.

Round 2

Reviewer 3 Report

I thank the authors for the work done and the correct changes made to the manuscript, you have greatly improved the article.

Only a few other minor changes are needed.

1. Check punctuation and delete empty spaces still present in the text.

2. Although most of the technical details of the products/equipment were provided, some are still missing. E.g.; tricaine (Line 168). Please add the company name and city for each product/equipment (please carefully see the Instructions for Authors of the journal or contact the editor to standardize this aspect).

3. For the C:N ratio of the control group, add the summary calculation in the supplementary material, stating in the main text that these values are hypothetical, given that the C:N ratio was not measured in this study.

4. Paragraph 2.1. "Ethics": please enrich the paragraph with what is written in the review response file: "Russell and Burch's 3Rs criteria were applied. Constant supervision of the behavior..."

5. Fish feed: please add the energy content of the feed 

Minor comments:

- Line 12: please replace "biofloc" with "biofloc technology system (BFT)" and thereafter use always BTF;

- Line 13: please replace "treatments" with "experimental conditions"

- Line 14: please replace "(control)" with "(control group)" and "BFT: (Biofloc)" with "BTF";

- Line 130: "2.5. Biofloc maduration" please correct;

- Line 138: please specify "TAN";

Minor editing of English language required

Author Response

Dear reviewer 3, thank you for taking your valuable time in reviewing the article one more time and giving us your thoughts.
Regarding the request, we inform you that we have reviewed again the punctuation and spaces in the text and we hope that this time everything has been left without unnecessary spaces and points.
The technical details of equipment and other products were added.
Information requested on ethics, C:N ratio and amount of energy in the food was also entered.
Other requested corrections were added to the text and highlighted in yellow. 
Best regards.